# Synergistic Effects of Subsoil Calcium in Conjunction with Nitrogen on the Root Growth and Yields of Maize and Soybeans in a Tropical Cropping System

**Murilo De Souza, Jéssica Pigatto de Queiroz Barcelos and Ciro A. Rosolem *** 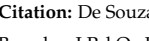

Department of Crop Science, College of Agricultural Sciences, São Paulo State University, Botucatu 18610-034, Brazil; jessica.pqb@gmail.com (J.P.d.Q.B.)
* Correspondence: ciro.rosolem@unesp.br

**Abstract:** A large part of Brazilian maize is double-cropped after soybeans, when water shortages are very frequent. A larger root system can mitigate drought stress and enable better nitrogen (N) use. Alleviating acidity and applying gypsum can increase root growth and N-use efficiency in maize, which has a more aggressive root system than soybeans. However, it is not known how these factors interact in integrated cropping systems, or how soybeans respond to them. Soybean and maize root growth and grain yields as affected by soil Ca enrichment using lime and gypsum, along with the N rates applied to maize intercropped with Guinea grass (*Megathyrsus maximus*), were assessed in a medium-term field experiment. Liming resulted in greater root growth for both crops; however, when lime was used in conjunction with gypsum, root growth was further enhanced. The total maize grain yield was 35% higher compare to the control when gypsum was used in conjunction with lime; however, subsoil Ca enrichment increased the total soybean grain yield by 8% compared to the control. Nitrogen fertilization increased the total maize grain yield by 36%, with a more expressive increase when applying 160 kg ha$^{-1}$ or more, and despite a positive effect on soybean grain yields in the long term, this response seems not to be a direct effect of the N applied to the maize. Both subsoil Ca enrichment and N application to maize increase root growth and the total yield of the system.

**Keywords:** gypsum; lime; Guinea grass; intercropping systems; acid soils



## 1. Introduction

Soil acidity affects approximately 30% of the world's potential food production area [1]. Acidic tropical soils are usually calcium (Ca)-deficient and show aluminum (Al) toxicity, which inhibits root growth and decreases agricultural production [2,3]. Surface liming has been effective in alleviating topsoil acidity; however, alleviating subsoil acidity by using lime alone is challenging due to its low solubility [4]. Agricultural gypsum—hydrated calcium sulfate (CaSO$_4$2H$_2$O)—has been used in conjunction with lime in acidic soils as an alternative to increase calcium (Ca) contents and alleviate Al toxicity in the subsoil. Some no-till studies have shown lime's effects in the subsoil even when applied on the soil surface, but this takes time [5,6], whereas the alleviation of subsoil acidity is faster with gypsum. Due to its higher mobility, gypsum application increases Ca$^{2+}$ and SO$_4^{2-}$ in the soil solution, facilitating leaching of these elements in the soil profile [7], and reduces aluminum activity and toxicity in the subsoil, favoring deep root growth [8]. A better root system results in higher soil exploration [9,10] and plays a crucial role in water acquisition and NUE [11–13] by avoiding N leaching [3]. In no-till systems, when lime is applied without incorporation, the pH is steeply increased close to the soil surface. At pH levels above 5.5, nitrification is enhanced, and the applied N is converted to nitrate, which can leach and take Ca$^{2+}$ with it, significantly improving the effect of lime [3]. Moreover, it has been shown that N fertiliza-

tion improves maize root branching and growth [14], which can further increase root growth in the soil profile.

Double-cropping of maize with forage grasses after soybeans has proven to be a sound and economical agricultural practice. However, as maize is grown after soybeans, and in many years this period is characterized by rainfall scarcity in tropical climate regions, there is a risk of decreased yields due to water stress. Therefore, improving subsoil conditions to establish a deep root system is paramount for the success of this cropping system [15,16]. Accordingly, greater crop responses to gypsum have been reported in water-deficient growing seasons [17,18]. Furthermore, a deeper root system results in increased N uptake by maize, higher N cycling within the system, and less N loss by leaching [3]. Forage grasses with vigorous root systems have been grown as cover crops or in association/consortium with maize in integrated systems in subtropical and tropical regions, and Guinea grass has been shown to be better than *Urochloa*, especially under N fertilization [19]. However, there is no reliable recommendation of N fertilization for maize/forage systems, especially when cropped after soybeans, since it is assumed that some of the atmospheric N fixed by soybeans could be available for maize. Therefore, considering the lower maize yield potential in this system, the higher N cycling, and the soybean contribution, the optimal N rate would be lower than when maize is grown as a lone crop.

Despite the general belief that mineral N application is unnecessary for inoculated soybeans, since the nutrient can be fully supplied by biological N fixation or by the soil [20], Salvagiotti et al. [21] reported that high-yield soybeans require large amounts of N to sustain their aboveground biomass and seeds with high protein content. These authors speculated that supplying N without decreasing the nodule activity could increase the soybean yield, and promising options include applying N before sowing or at depths below the nodulation zone—that is, increasing the availability of N in the soil profile. However, this has been not demonstrated so far.

Therefore, the hypothesis is that subsoil Ca enrichment with lime and gypsum, in a production system with maize double-cropped with Guinea grass after soybeans, will increase root growth in the soil profile, resulting in higher yields of maize and soybeans, while better root growth of the grasses along with N fertilization and higher dry matter production can also improve soybeans' root growth and grain production. Although the effect of Ca in the subsoil increasing root growth is not new, there is a gap in knowledge about its legacy effect for the next crop, and its interaction with N in integrated cropping systems with maize cropped after soybeans has not yet been addressed. The objective of this study was to evaluate the effects of enriching the subsoil with Ca using lime alone or in conjunction with gypsum and N fertilizer on root growth and grain yields in a no-till cropping system.

## 2. Materials and Methods

### 2.1. Site Description

The experiment was carried out in Botucatu, São Paulo State, Brazil, from 2016 to 2020, in a clayey, kaolinitic, thermic Typic Haplorthox [22] located at 48°25′38.84″ W, 22°49′50.90″ S, 790 m above sea level. The climate is Cwa, i.e., tropical with dry winters and warm, rainy summers. Precipitation and temperatures were recorded during the experiment at a meteorological station located 300 m from the experimental area (Figure 1).

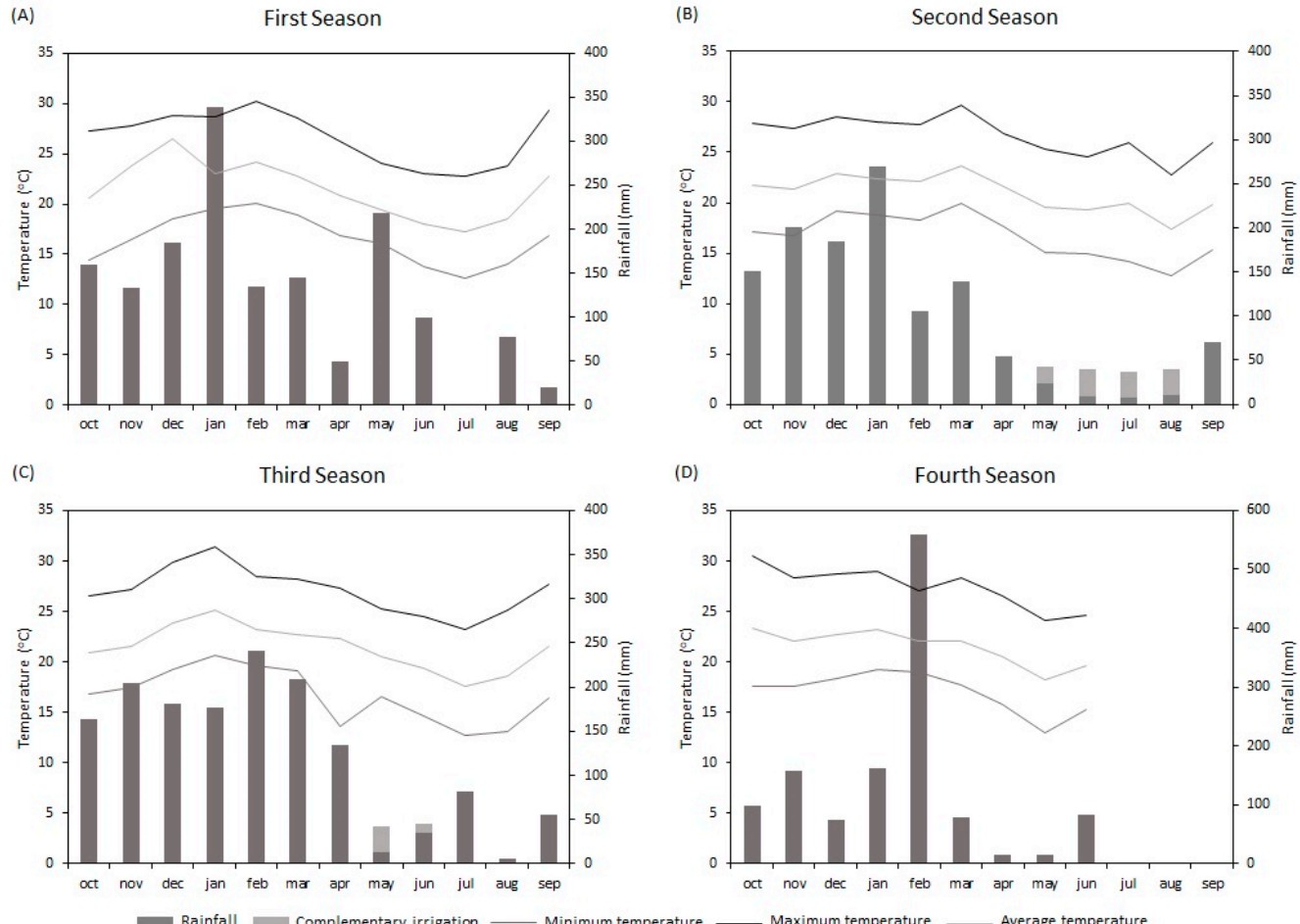

**Figure 1.** Rainfall and minimum, average, and maximum temperatures in the first (**A**), second (**B**), third (**C**), and fourth (**D**) growing seasons: 2016/2017, 2017/2018, 2018/2019, and 2019/2020, respectively. Botucatu, SP.

Before the experiment, the area was under fallow, with a mix of grasses and some broad leaves. In September 2016, the soil was sampled for chemical characterization analyses [23], and the results are shown in Table 1.

**Table 1.** Soil chemical characteristics at four depths in the experimental area before initiating the experiment, August 2016.

| Depth | pH [#] | OM [†] | P | S | K | Ca | Mg | Al | H + Al | CEC [‡] | BS [§] |
|---|---|---|---|---|---|---|---|---|---|---|---|
| m | $CaCl_2$ | g dm$^{-3}$ | mg dm$^{-3}$ | | -------------- mmol$_c$ dm$^{-3}$ -------------- | | | | | | % |
| 0.00–0.10 | 4.4 | 18 | 13 | 5 | 2.1 | 12 | 12 | 9 | 52 | 78 | 32 |
| 0.10–0.20 | 4.3 | 14 | 10 | 5 | 1.1 | 9 | 8 | 9 | 48 | 66 | 27 |
| 0.20–0.40 | 4.1 | 11 | 6 | 22 | 0.5 | 4 | 4 | 12 | 55 | 63 | 13 |
| 0.40–0.60 | 4.0 | 11 | 6 | 21 | 0.4 | 3 | 4 | 20 | 90 | 97 | 8 |

[#] Soil pH measured in calcium chloride solution. [†] Organic matter. [‡] Cation-exchange capacity. [§] Base saturation.

### 2.2. Experimental Design and Treatments

The treatments were lime, lime + gypsum, and control, with 0, 80, 160, and 240 kg ha$^{-1}$ of N applied to the maize, arranged in a 3 × 4 factorial scheme in completely randomized blocks with four replications. The lime ($CaCO_3$) rate was calculated to raise the soil's base saturation to 70%, and the gypsum ($CaSO_4 2H_2O$) rate was calculated using the average soil clay content from 0 to 0.4 m multiplied by six, as recommended by Duarte et al. (2022).

The rates corresponding to 2.92 Mg ha$^{-1}$ lime and 2.0 Mg ha$^{-1}$ gypsum were applied on the soil surface. Gypsum was applied to the respective plots immediately after the application of lime. In September 2017, lime and gypsum were reapplied at the same rates. The four rates of N were applied annually to maize intercropped with Guinea grass *(Megathyrsus maximus)*.

### 2.3. Experiment Management

The plots consisted of 10 soybean or maize rows that were 10 m long and spaced 0.45 m apart from one another. The spontaneous vegetation was desiccated using glyphosate (2.30 kg$^{-1}$ a.i.). The lime + gypsum treatment was applied in September 2016 and October 2017. Soybean, cv. TMG 7062 IPRO (Tropical Improvement and Genetics), was planted each year in November, over the desiccated residues of the previous spontaneous species or maize/Guinea grass, which was desiccated two weeks before soybean planting each year. The seeds were inoculated with *Bradyrhizobium japonicum*. Phosphorus (P) and potassium (K) were applied in the seed furrow at 26 kg ha$^{-1}$ and 50 kg ha$^{-1}$, respectively, each year, as triple superphosphate [Ca(H$_2$PO$_4$)$_2$] and potassium chloride (KCl). Soybean was harvested 125, 145, 136, and 128 days after emergence in the growing seasons 2017, 2018, 2019, and 2020, respectively, and the grain yield was adjusted to 13% moisture.

Maize (Dow AgroSciences Hybrid 2B 587 RRBTPW) was planted after the soybean harvest each year, with a population of 55,000 plants ha$^{-1}$, intercropped with Guinea grass using 10 kg ha$^{-1}$ of pure live seeds. The forage seeds were mixed with the phosphate fertilizer and applied at a depth of 0.08 m. Each plot received 35 kg ha$^{-1}$ P as triple superphosphate. Potassium chloride was used to supply K at 82 kg ha$^{-1}$ at sowing, plus 41 kg ha$^{-1}$ at V4 (i.e., plants with four fully developed leaves). Ammonium sulfate [(NH$_4$)$_2$SO$_4$] was applied to supply N at 30 kg ha$^{-1}$ at sowing, completed with 50, 130, and 210 kg ha$^{-1}$ side-dressed 0.1 m from the plant line to the respective treatments at stage V4. For a yield of 6–8 Mg ha$^{-1}$, 120 kg ha$^{-1}$ N would be recommended [24]. The rates used in this experiment ranged from low to very high, because we wanted to know the effects on the subsequent soybean crop. Maize was harvested at, 155, 122, and 132 days after plant emergence in the growing seasons 2017, 2018, 2019 and 2019, respectively. The grain moisture was corrected to 13%.

### 2.4. Root Sampling and Dry Matter Determination

Root samples were collected at the depths 0 to 0.1, 0.1 to 0.2, 0.2 to 0.4, and 0.4 to 0.6 m, using a steel probe with a 0.075 m internal diameter. Five root subsamples were randomly taken per plot for soybeans and maize, in the planting row and between rows. For soybeans, sampling was performed at R2—the full flowering stage [25]—on 24 January 2017, 8 January 2018, 15 January 2019, and 13 January 2020. For maize intercropped with Guinea grass, the samples were taken on 22 June 2017, 29 July 2018, and 7 July 2019. The roots were carefully separated from the soil and other residues by washing them under a flow of swirling water over a 0.5 mm mesh sieve. Then, the roots were immersed in 30% ethyl alcohol solution, placed in plastic pots, and stored under refrigeration at 2 °C. Afterward, the roots were scanned [26] using an optical scanner (Scanjet 4C/T, HP) at 300 dpi resolution and analyzed with WinRHIZO version 3.8-b (Regent Instrument Inc., Quebec, QC, Canada). The samples were dried in a forced-air oven (Fanen, model 32 E, Brazil) at 60 °C for 48 h to assess the roots' dry matter.

### 2.5. Soil Sampling and Chemical Analysis

Soil samples were collected 12, 24, and 36 months after the first lime application, in September 2017, 2018, and 2019, respectively, up to the depth of 0.6 m. Four subsamples were randomly collected and combined into a composite sample, and exchangeable Ca was extracted with ion-exchange resin [23].

*2.6. Statistical Analysis*

Data from each year and each soil layer were analyzed separately. After testing for normality and homoscedasticity, the root length density, dry matter, and grain yield data were subjected to ANOVA. Blocks were considered as random effects, and for the first soybean crop, one-way ANOVA was used. For the remaining years, a factorial ANOVA was used based on a completely randomized block design with two factors (corrective and nitrogen rates). When the ANOVA result was significant, the modified *t*-test (Fisher's protected least significant difference (LSD) at $p \leq 0.05$) was used to separate the means. SAS software, version 9.4, was used. Pearson's correlation coefficients between maize and soybean root length densities were determined ($p \leq 0.05$).

## 3. Results

Soil $Ca^{2+}$ was increased by liming in the second, third, and fourth growing seasons up to the depth of 0.60 m (Figure 2A–C). However, when gypsum was used in conjunction with lime, the $Ca^{2+}$ concentrations were higher in the soil profile. Nitrogen application further increased the percolation of $Ca^{2+}$ through the soil profile, with rates over 160 kg ha$^{-1}$, but in the fourth growing season there was generally no significant effect of N fertilization (Figure 2F).

There were no significant interactions of lime application with N rates for root length density (RLD), root dry matter (RDM), soil $Ca^{2+}$ concentrations, or grain yield for any of the crops and in any of the growing seasons. In the upper soil layers, RLD was generally higher—both in the soybean plant rows and between rows—with lime or lime + gypsum compared to the control (Figure 3), except in the first growing season (2016/2017) in the 0.10 to 0.20 m layer (Figure 3(A1)). In the third growing season, the RLD was higher when lime was used in conjunction with gypsum compared with isolated lime—both in the soybean plant rows and between rows—in the 0.00 to 0.10 m layer (Figure 3(E1,F1)). In the fourth growing season (2019/2020), a higher RLD was also observed between soybean rows in the 0.10 to 0.20 m layer when lime was applied in conjunction with gypsum, with values higher than the other treatments (Figure 3(H1)). In the subsoil (0.40–0.60 m layer), the application of lime in conjunction with gypsum increased the RLD compared with the control, except for between soybean rows in the first growing season (Figure 3(B1)). In the third and fourth growing seasons, there was no difference between lime and lime + gypsum in the soybean plant rows (Figure 3(E1,G1)). However, in the fourth growing season, the RLD between soybean rows was higher when lime was used in conjunction with gypsum (Figure 3(H1)). The soybean RDM was higher in almost the entire soil profile after alleviating acidity (Figure 3). However, in the 0.00 to 0.10 m layer, there was no difference in the plant rows in the second growing season (2017–2018) or between rows in the second, third, and fourth growing seasons. Furthermore, lime + gypsum resulted in higher RDM only in the third growing season at the 0.10 to 0.20 m layer.

The use of lime in conjunction with gypsum resulted in higher RDM of soybeans in the subsoil compared with the control (Figure 3), regardless of the sampling location. Comparing lime + gypsum with lime, there was an effect on RDM only in the plant rows in the 0.20 to 0.40 m layer in the second and third growing seasons (Figure 3(C2,E2)), and in the 0.40 to 0.60 m layer in the second and fourth growing seasons (Figure 3(C2,G2)). However, between rows, the RDM was higher in the 0.20 to 0.40 m layer in the first, third, and fourth growing seasons (Figure 3(B2,F2,H2)), and in the 0.40 to 0.60 m layer in the second, third, and fourth growing seasons (Figure 3(D2,F2,H2)). The differences found in the results for the plant rows and between rows probably occurred due to the water deficit during the experiment (Figure 1).

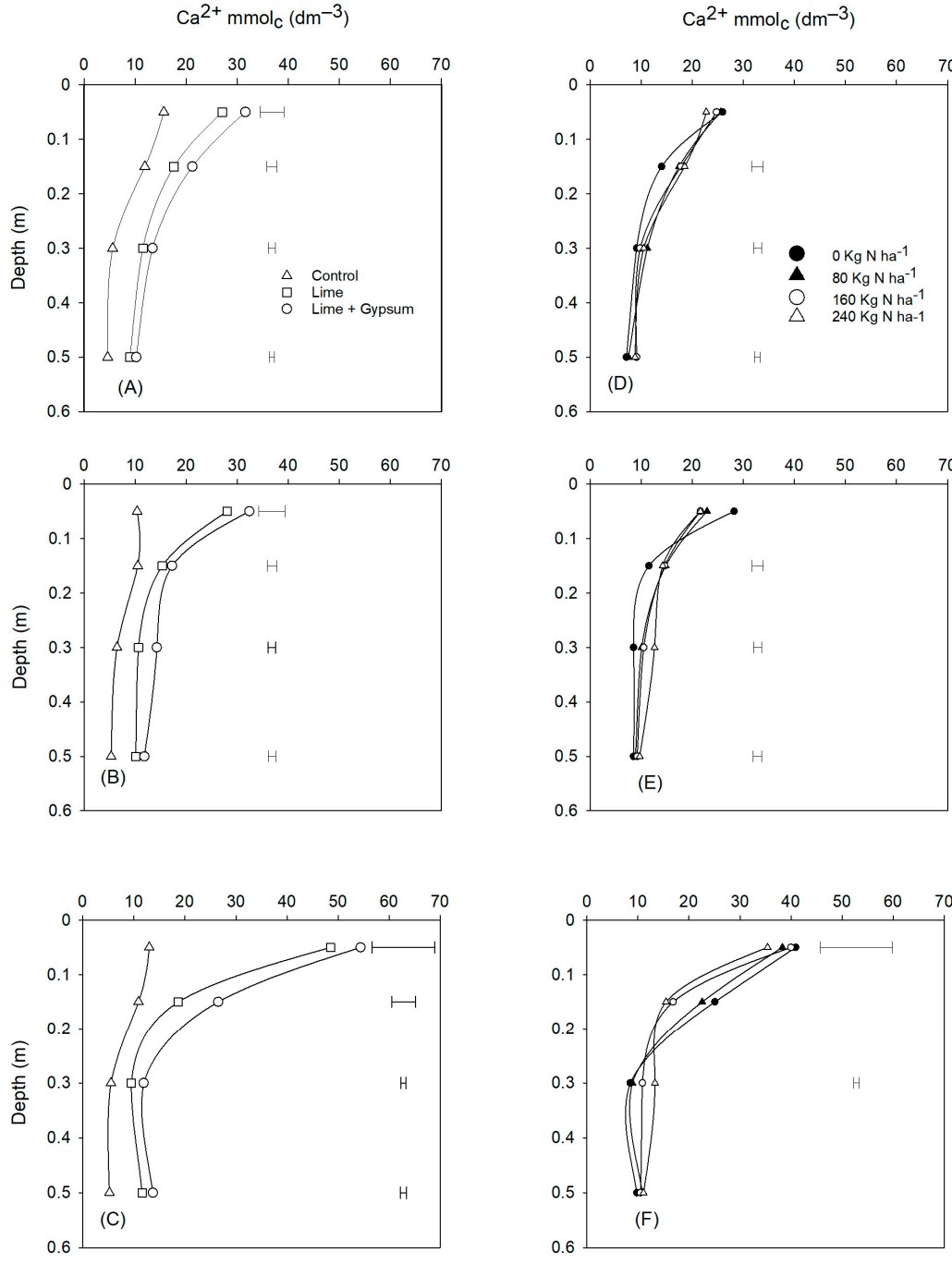

**Figure 2.** Calcium (Ca) concentration in the soil as affected by lime and lime + gypsum in the growing seasons 2017 (**A**), 2018 (**B**), and 2019 (**C**), and N rates in the growing seasons 2017 (**D**), 2018 (**E**), and 2019 (**F**).

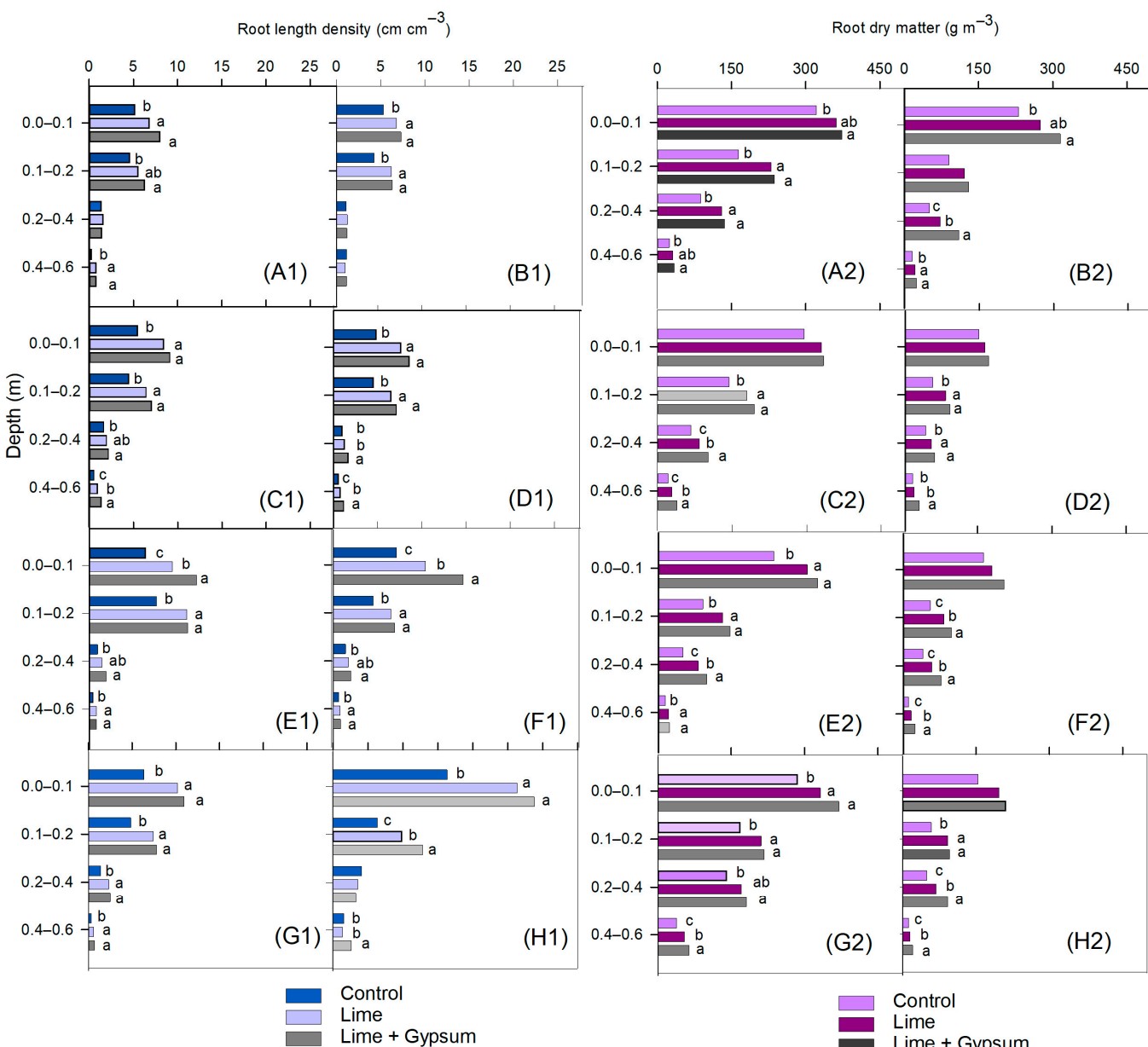

**Figure 3.** Soybean root length density within rows—growing seasons 2016/2017 (**A1**), 2017/2018 (**C1**), 2018/2019 (**E1**), and 2019/2020 (**G1**)—and between rows—growing seasons 2016/2017 (**B1**), 2017/2018 (**D1**), 2018/2019 (**F1**), and 2019/2020 (**H1**)—and soybean root dry matter in rows—growing seasons 2016/2017 (**A2**), 2017/2018 (**C2**), 2018/2019 (**E2**), and 2019/2020 (**G2**)—and between rows—growing seasons 2016/2017 (**B2**), 2017/2018 (**D2**), 2018/2019 (**F2**), and 2019/2020 (**H2**)—as affected by lime and gypsum application. Different letters indicate means that are statistically different according to the Turkey HSD test ($p \leq 0.05$).

When N was applied to maize at 160 and 240 kg ha$^{-1}$ N, the soybean RLD was higher in the plant rows at the 0.00 to 0.10 m soil layer and in the subsoil (0.40–0.60 m) in the second soybean growing season (Figure 4(A1)). Between the soybean rows, the RLD was higher with 240 kg ha$^{-1}$ N up to 0.20 m, and in the subsoil the highest soybean RLD was observed in the treatments receiving 160 and 240 kg ha$^{-1}$ N (Figure 4(B1)). In the third growing season, a higher RLD was observed in the surface layer and the 0.40 to 0.60 m subsoil layer under the plant rows when soybeans were grown after N-fertilized maize, regardless of the applied rate (Figure 4(C1)). However, between the soybean rows (Figure 4(D1)), the RLD was higher in the treatments receiving N at 160 and 240 kg ha$^{-1}$,

except in the 0.00 to 0.10 m layer, where the increase was observed only with 240 kg ha$^{-1}$ compared with the control. In the fourth growing season, the soybean RLD was increased in the plant rows by N fertilization up to 160 kg ha$^{-1}$, only in the 0.20 to 0.40 m layer (Figure 4(E1)). The soybean RDM was higher in the plant rows with the use of higher N rates compared with the control in the uppermost soil layer (Figure 4(A2)). In the subsoil (0.20–0.40 layer), the RDM was higher in the treatments with N in the second (Figure 4(A2)) and third (Figure 4(C2)) growing seasons compared with the control. In the 0.40 to 0.60 m layer, both in the plant rows and between rows, N rates of 160 and 240 Kg ha$^{-1}$ resulted in higher RDM compared with the treatment without N in the second, third, and fourth growing seasons (Figure 4(A2–C2,E2,F2)), except between rows in the third growing season (Figure 4(D2)).

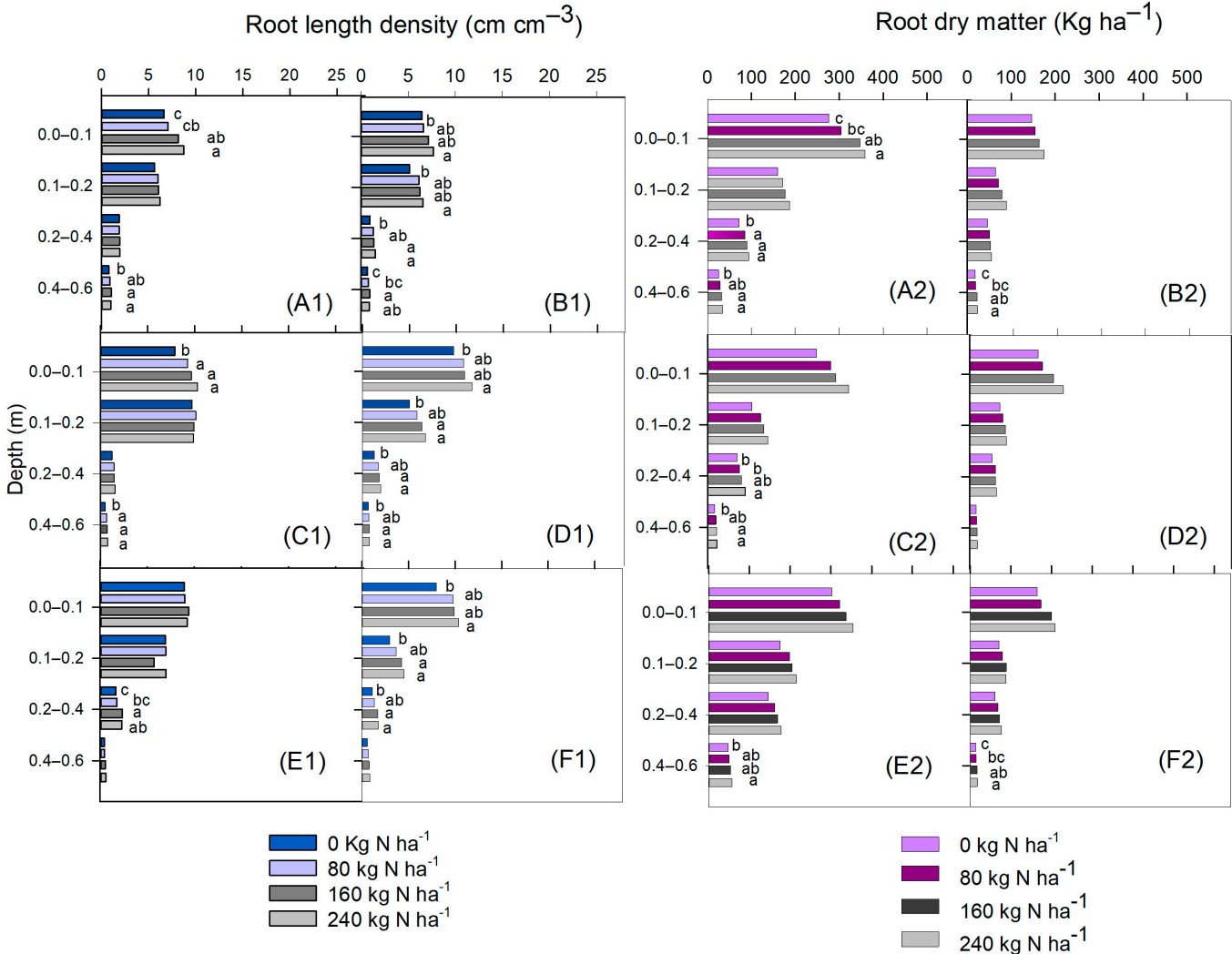

**Figure 4.** Soybean root length density within plant rows—growing seasons 2017/2018 (**A1**), 2018/2019 (**C1**), and 2019/2020 (**E1**)—and between rows—growing seasons 2017/2018 (**B1**), 2018/2019 (**D1**), and 2019/2020 (**F1**)—and soybean root dry matter in rows—growing seasons 2017/2018 (**A2**), 2018/2019 (**C2**), and 2019/2020 (**E2**)—and between rows—growing seasons 2017/2018 (**B2**), 2018/2019 (**D2**), and 2019/2020 (**F2**)—as affected by N rates. Different letters indicate means that are statistically different according to the Tukey HSD test ($p \leq 0.05$).

The RLD in the maize/Guinea grass intercropping was increased by lime (whether used in conjunction with gypsum or not) in all growing seasons, both in the plant rows and between rows, with a few exceptions (Figure 5). In the second and third growing seasons, the RLD was higher from 0.20 to 0.60 m (Figure 5(C1–F1)) when lime was used in

conjunction with gypsum. In general, it was higher with the use of gypsum in the subsoil than when lime was applied alone. However, generally, there was no difference in the RDM when lime was applied with or without gypsum (Figure 5(A2–F2)). The response of RLD to N in the maize/Guinea grass consortium was significant up to 240 kg ha$^{-1}$ in almost all soil layers (Figure 6). However, N application did not increase the RDM in the plant rows (Figure 6), except for the 0.40 to 0.60 m layer in the first growing season (Figure 6(A2)) and the 0.20 to 0.40 m layer in the second growing season (Figure 6(C2)). Between rows, an increase in RDM was observed only in the second growing season in the 0.40 to 0.60 m layer for the highest rate (Figure 6(D2)), and in the third growing season in the 0.00 to 0.10 m layer for all N rates, compared with the control (Figure 6(F2)).

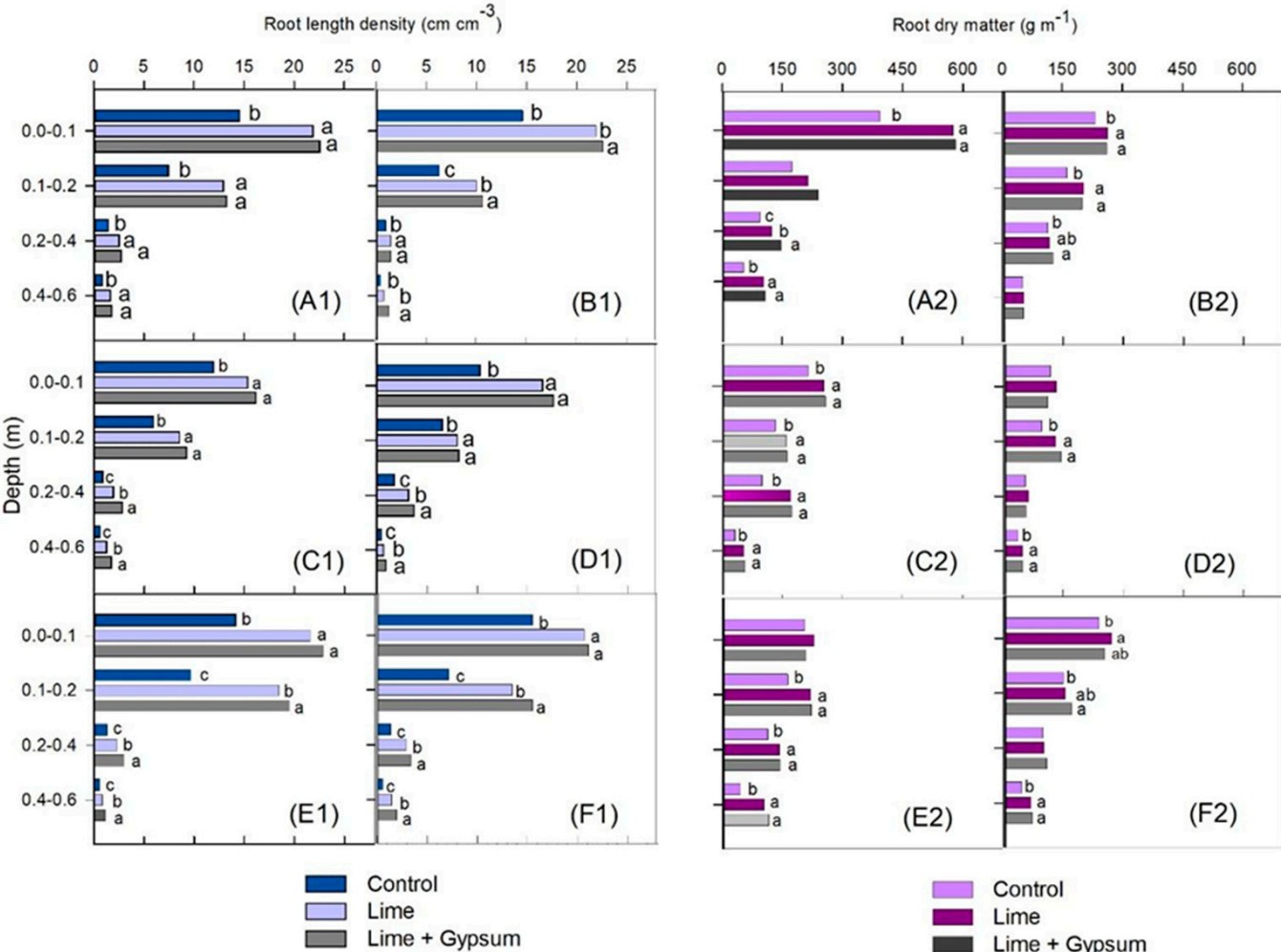

**Figure 5.** Root length density of maize intercropped with Guinea grass within plant rows—growing seasons 2017 (**A1**), 2018 (**C1**), and 2019 (**E1**)—and between rows—growing seasons 2017 (**B1**), 2018 (**D1**), and 2019 (**F1**)—and root dry matter in rows—growing seasons 2017 (**A2**), 2018 (**C2**), and 2019 (**E2**)—and between rows—growing seasons 2017 (**B2**), 2018 (**D2**), and 2019 (**F2**)—as affected by lime and gypsum application. Different letters indicate means that are statistically different according to the Tukey HSD test ($p \leq 0.05$).

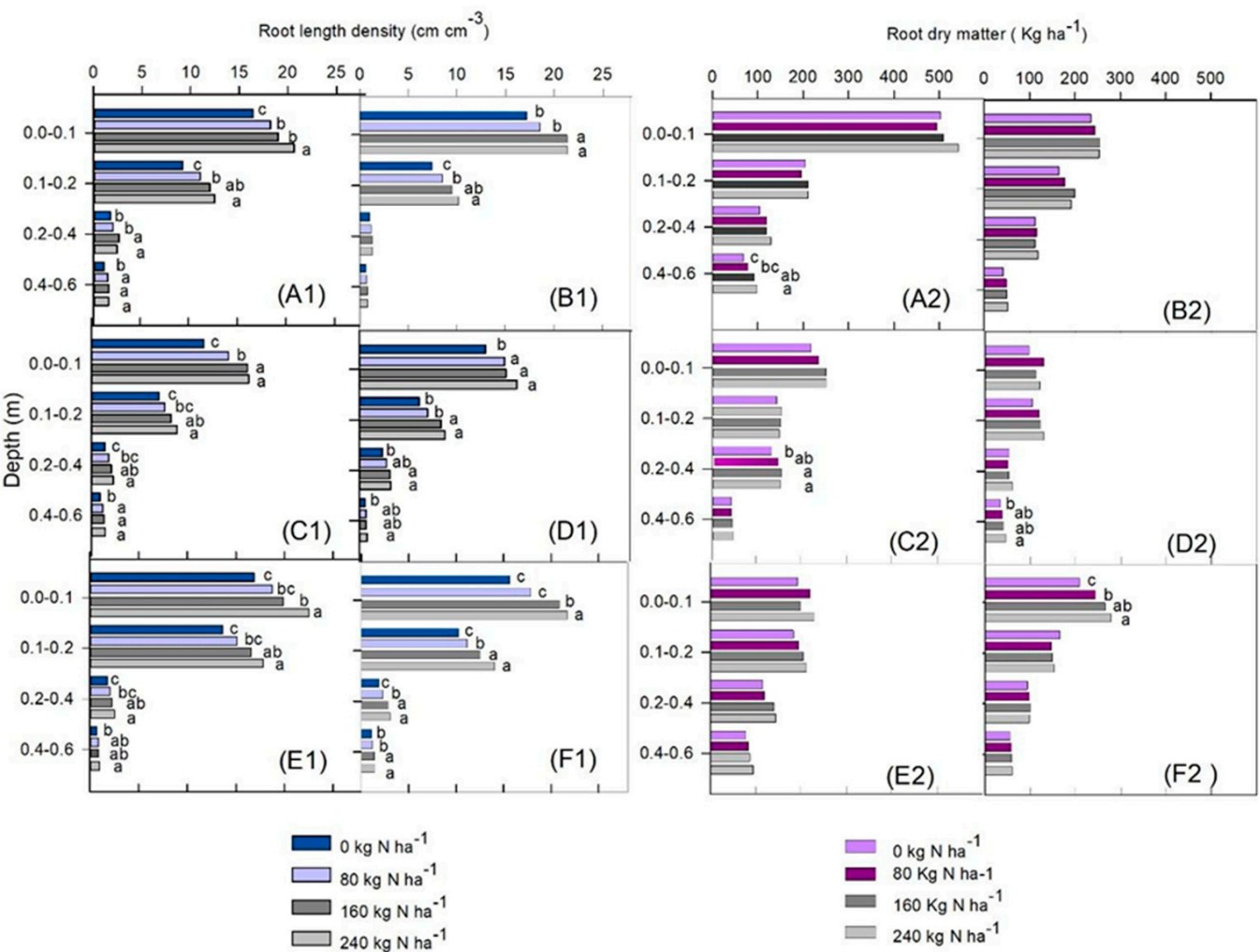

**Figure 6.** Root length density of maize intercropped with Guinea grass within plant rows—growing seasons 2017 (**A1**), 2018 (**C1**), and 2019 (**E1**)—and between rows—growing seasons 2017 (**B1**), 2018 (**D1**), and 2019 (**F1**)—and root dry matter in rows—growing seasons 2017 (**A2**), 2018 (**C2**), and 2019 (**E2**)—and between rows—growing seasons 2017 (**B2**), 2018 (**D2**), and 2019 (**F2**)—as affected by N rates. Different letters indicate means that are statistically different according to the Tukey HSD test ($p \leq 0.05$).

It is interesting to observe that the soybean RLD was correlated with the RLD of the intercrop in all seasons and soil depths, with a few exceptions (Table 2).

**Table 2.** Correlation coefficients of root length density between maize intercropped with Guinea grass and soybeans (plant rows and between rows) in growing seasons 2017/2018, 2018/2019, and 2019/2020.

| Depth (m) | 2017/2018 | | 2018/2019 | | 2019/2020 | |
|---|---|---|---|---|---|---|
| | Row | Inter-Row | Row | Inter-Row | Row | Inter-Row |
| 0–0.10 | 0.641 ** | 0.588 ** | 0.733 ** | 0.686 ** | 0.493 ** | 0.597 ** |
| 0.10–0.20 | 0.481 ** | 0.630 ** | 0.319 * | 0.544 ** | 0.407 ** | 0.661 ** |
| 0.20–0.40 | 0.292 * | 0.269 ns | 0.433 ** | 0.302 * | 0.565 ** | −0.066 ns |
| 0.40–0.60 | 0.699 ** | 0.732 ** | 0.735 ** | 0.448 ** | 0.387 ** | 0.462 ** |

* Significant $p < 0.05$. ** Significant $p < 0.01$. ns means not significant.

When the effects of lime and gypsum were compared year by year, there was no difference in soybean grain yield up to the third growing season. However, in the fourth growing season, lime increased the grain yield by 10.8% compared with the control, and no further increase was observed when used in conjunction with gypsum. However, when looking at the accumulated soybean grain yield, the response to lime was also 5.8% higher than in the control (Figure 7A).

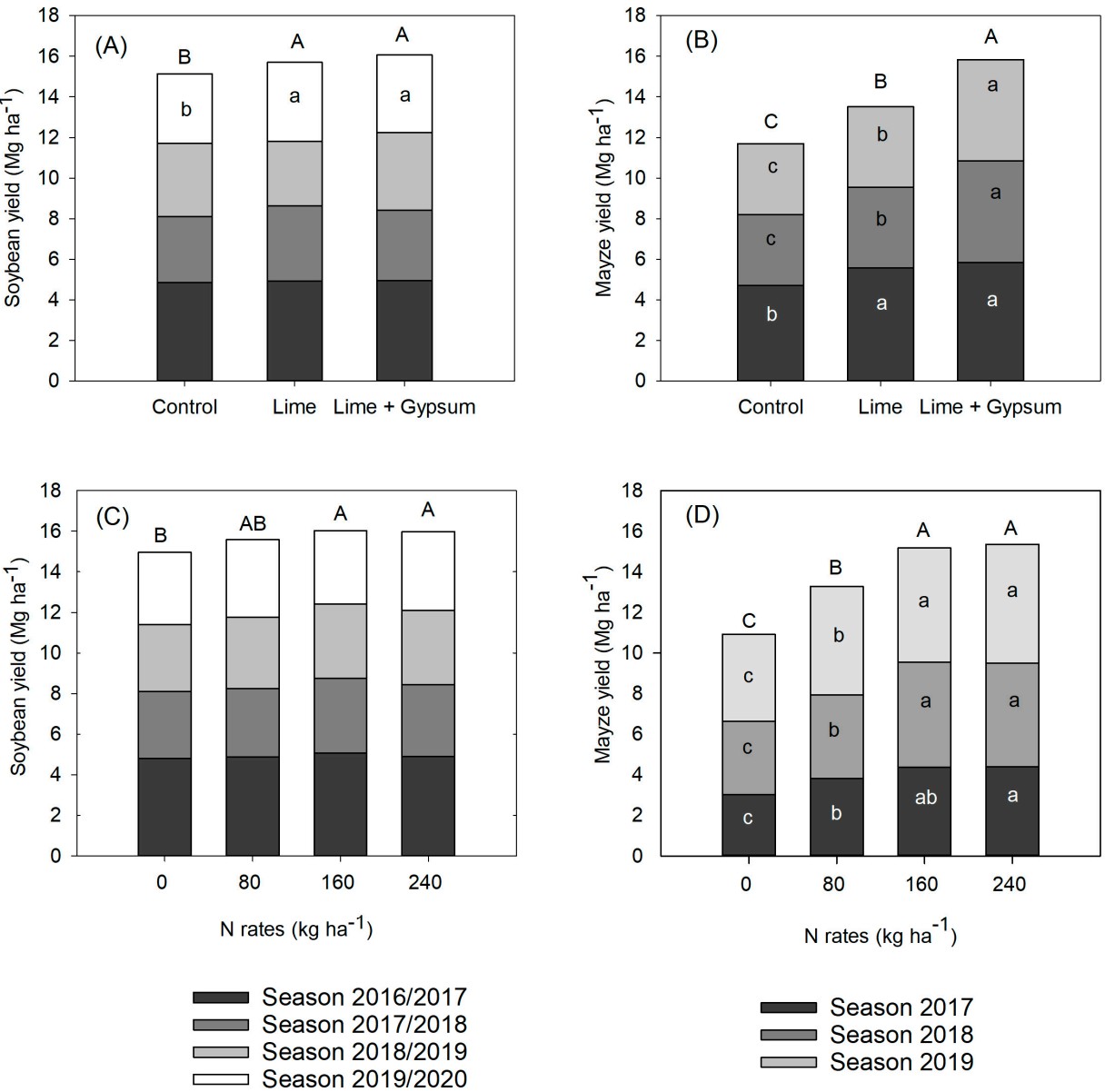

**Figure 7.** Soybean and maize yields as affected by lime and lime + gypsum (**A** and **B**, respectively) and by nitrogen rates (**C** and **D**, respectively). Means followed by a common letter are not significantly different between amendments or N rates (LSD, $p < 0.05$).

Again, there was no effect of N rates on soybean grain yield when analyzed year by year. The N concentrations in soybean leaves were not affected by the treatments and averaged 52.5 mg kg$^{-1}$ over four years. However, the accumulated grain yield was higher with 160 and 240 kg ha$^{-1}$ applied to the consortium compared with the treatment without N (Figure 7C). It is important to note that in the first soybean crop, the effect of the N dose was not considered, since nitrogen was applied only to the maize crop.

Maize was more responsive than soybeans to liming (Figure 7B). In 2017, the application of lime and lime + gypsum resulted in an average increase of 17.8% compared with the control. However, in 2018, the combination of gypsum with lime resulted in higher maize grain yields: 20.5% and 29.8% higher compared to lime alone and the control, respectively. In 2019, again, the addition of gypsum increased the grain yield by 7.2% compared with lime alone and 20.2% compared with the control, and these increases were reflected in the accumulated grain yields (Figure 7B). However, there was no significant effect of lime or gypsum on maize leaf N concentrations.

Maize grown in association with Guinea grass as a relay crop after soybeans responded to N fertilization up to 160 kg ha$^{-1}$ (Figure 7B), and the leaf N content was increased from 28.7 mg kg$^{-1}$ to 34.2 mg kg$^{1}$ on average. Although the N rate of 80 kg ha$^{-1}$ increased the yield by 2988 kg ha$^{-1}$ compared with the control, when 160 and 240 kg ha$^{-1}$ were applied there was an average increase of 5104 kg ha$^{-1}$ in the accumulated grain yields.

## 4. Discussion

The enrichment of the soil profile with $Ca^{2+}$ after the application of gypsum was expected; however, this has been seldom observed when lime is applied alone [4,27]. Gypsum is considered to be an important alternative to improve root system distribution in the soil profile [3], and its association with lime is efficient in improving the $Ca_2^+$ content in the soil.

As we hypothesized, $Ca^{2+}$ leaching to the subsoil was improved by N fertilization (Figure 2D–F). This result can be explained, since when lime is applied to the soil surface there is a sharp pH increase close to the surface, and the N from the fertilizer is transformed into nitrate. According to Rosolem et al. [28], nitrate is mobile in the soil profile and tends to follow the water infiltration flow, which is facilitated under no-till conditions due to both less evaporation and better soil profile structuring. Pearson et al. [29] reported that nitrogen fertilization increased the movement of $Ca^{2+}$ along the soil profile when acidity amendments were applied. The authors attributed this effect to the formation of soluble salts such as Ca $(NO_3)_2$, subject to leaching by the downward movement of water. Additionally, the movement of small particles of lime [4] may have played a role in increasing Ca throughout the soil profile.

Root growth was positively influenced by soil acidity alleviation (Figure 3). Calcium is essential for root elongation [30], so root development is impaired in acidic soils or under low concentrations of $Ca^{2+}$, especially in the subsoil [3]. There are reports of regular growth of soybean roots with 10 mmolc dm$^{-3}$ $Ca^{2+}$ [31], and the response is expected to be low when the soil concentrations are over 12 mmolc dm$^{-3}$ [32], but the response was significant with higher $Ca^{2+}$ contents in the deeper soil layers in this experiment. This could have been due to the weather conditions, in response to dry spells. In the present study, although lime was found to be efficient in increasing the soybean root system, greater development was observed in the subsoil when it was used in conjunction with gypsum (Table 1), probably due to the higher $Ca^{2+}$ content at this depth (Figure 2). Despite a report of greater soybean root growth when the soil $Ca^{2+}$ content was increased after the application of gypsum, the effect of Ca on root growth after soil acidity correction was inconsistent for soybeans.

The effect of soil and subsoil acidity alleviation on root growth in the intercropped maize/Guinea grass was more evident than that observed in soybeans after liming, with or without gypsum. This suggests that maize is more responsive to soil acidity correction. Variations in responses between different plant species occur because of genetic factors linked to the plants' efficiency in acquiring soil $Ca^{2+}$, since this only occurs in new, non-suberized parts of the roots, and the availability of $Ca^{2+}$ in situ is paramount for constant absorption by very young roots. Therefore, the better root distribution in the subsoil observed in the present study could be related to the $Ca^{2+}$ concentrations, since the application of gypsum in conjunction with N fertilization was efficient in increasing its contents in the subsoil compared to those observed with the application of lime alone. On average, the $Ca^{2+}$ content in the subsoil was increased in this experiment (Figure 2), reaching up to

20.0 mmolc dm$^{-3}$ (0.20–0.60 m layer). However, there is evidence of maize root responses to Ca$^{2+}$ in the order of 15 mmolc dm$^{-3}$ [33]. In addition, an effect on root growth was observed throughout the soil profile after N supply (Figures 4 and 6). Nitrate (NO$_3^-$) is known to play a signal-regulating role in many physiological processes, including root growth, and an interconnection of the concentrations of NO$_3^-$ and Ca$^{2+}$ [34,35] with auxin is possible [36]. These processes are controlled by several concentrations of gene transcription, which are regulated by NO$_3^-$ [37]. Studies have shown that the growth of the root system is regulated by NO$_3^-$ and auxin signaling pathways [38,39].

It has been demonstrated [9] that intercropping systems with forage grasses are efficient in the use of N, avoiding N leaching, and it has been reported that Guinea grass is highly demanding with respect to N [40]. In addition, deep root growth may also have been favored by the presence of Guinea grass, which has an aggressive root system and was alive up to the desiccation before the next soybean crop. Soybeans' root growth is improved as a result of the previous root growth of cover crops [41], probably because of the biopores present in the soil profile. Under no-till conditions, without soil disturbance, continuous channels are formed by decomposing roots, which serve as paths that favor the root growth of subsequent crops in the soil profile [42,43]. This is supported by the positive correlation between soybean root growth and the root length density of the intercrop (Table 2). As *Urochloa* species have a vigorous, abundant, and deep root system, these plants can explore a large volume of soil and take up greater amounts of nutrients available in soil regions that are far from the roots of the consortium's grain-producing crop, which are usually more superficial and sparser [3]. In addition, its aggressive root system improves the soil's physical condition by increasing its pore continuity [9], which may result in greater soil microporosity, and the root length density is improved at higher soil microporosity [44].

*Grain Yield*

Soybean yields were higher when soil acidity was corrected, with significant differences for the last season and the accumulated yields. Several studies have shown no soybean yield responses to the superficial application of lime and/or gypsum when there was no water shortage [45,46]; thus, the lack of response was attributed to the adequate rainfall conditions during crop development. In addition, greater organic matter and nutrient accumulation on the soil surface under NT decreases Al toxicity through the formation of Al–organic complexes [47]. This may explain the lack of soybean response in the first three harvests of this study, since there was no water deficit in this period (Figure 1). However, the 2018/2019 season was marked by low rainfall (total precipitation of 493 mm) and, eventually, the soybean yield was lower in the control treatment (3.4 Mg ha$^{-1}$) than after the application of lime (3.9 Mg ha$^{-1}$) and lime + gypsum (3.8 Mg ha$^{-1}$), corroborating previous observations. The increase in grain yield may have been associated with the increased concentrations of Ca$^{2+}$ in the soil [17]. The authors observed responses in soybean grain yield when rainfall conditions were unfavorable, applying a similar rate of gypsum to that used in the present study. Thus, the increase in soil Ca$^{2+}$ concentration (Figure 6) with the application of lime and gypsum improved root development (Figure 3), which lessened the effects of water shortage on plant growth and yield. This study demonstrates that soil acidity alleviation is efficient in increasing soybean yields throughout the seasons, justifying the need for correction even when year-to-year comparisons do not show significant differences.

The application of N rates of 160 and 240 kg ha$^{-1}$ to maize intercropped with Guinea grass positively affected the accumulated soybean grain yields (Figure 7). Biological nitrogen fixation (BNF) can supply the N requirements of soybeans through the use of adapted rhizobia strains selected for tropical conditions [20]. However, there are studies reporting that the decrease in BNF generally observed after flowering may restrict N's availability to soybeans, which would not be able to acquire adequate amounts of the nutrient for high yields. Salvagiotti et al. [21] analyzed several field studies in different regions after a comprehensive literature review and observed that high-yield soybeans

require large amounts of N to support their aboveground biomass and protein-rich seeds. According to the authors, the supply of N without decreasing the nodule activity could increase crop yield and, as promising options, they suggested supplying N before sowing or at depths below the nodulation zone—that is, increasing N's availability in the soil profile. Despite the efficiency of N supply by BNF for soybeans in tropical soils [20], there was a response to the N applied to the previous maize. Crop and cover crop residues accumulated on the soil surface constitute an important nutrient reserve, whose availability can be fast and intense [3] or slow and gradual, depending on precipitation, temperature, soil microbial activity, and the quality and quantity of plant residue [48]. Additionally, Guinea grass has a high N-cycling capacity in the system; therefore, the amount of N recycled by Guinea grass certainly increased the total N availability as the system progressed. However, in this study, the N concentration in soybean leaves was not affected by the application of N to maize. Therefore, the effect of maize fertilization on soybean yields was not a direct effect of the nutrient but, rather, a consequence of a general improvement of the system.

In the first season of maize intercropped with Guinea grass, the grain yield was increased by liming, with no further increase when gypsum was applied (Figure 7). Favorable rainfall conditions in this season (Figure 1) can explain this result, as a greater crop response to gypsum has been reported [17] when there was a water deficit. In the second and third seasons of maize intercropped with Guinea grass, lime + gypsum resulted in a higher yield when compared with lime alone. In addition, the accumulated maize grain yield was higher when gypsum was used in conjunction with lime (Figure 7). These results confirm that the use of gypsum is an important tool to increase maize yields when cropped after soybeans in a period in which there is usually a decrease in soil water availability that results in plant drought stress. Caires et al. [46] also observed significant increases in maize yields due to the higher Ca availability in deep soil, and the authors related these results to a better distribution of the crop's root system. A long-term experiment with maize concluded that the combined application of gypsum and lime resulted in a 17% increase in yields [4], highlighting the importance of this tool in maximizing the grain yield of this species under water shortage during crop development. Penariol et al. [49] showed that maize's grain yield can be compromised if there is a water deficit during flowering—a phase that determines the number of ovules to be fertilized and, consequently, grain production. This explains the lower yields observed in the 2018 season compared with 2017 and 2019.

In general, an increase in maize grain yield was observed when N was added after soil acidity alleviation compared with the control treatment, even when a low rate (80 kg ha$^{-1}$) was applied. These results suggest a response of maize grown as a relay crop to lower N doses, which can be explained by the high N residues deposited on the soil surface by soybeans, plus the N recycled by the grass. There is a fast decomposition of soybean residues due to the low C/N ratio, which benefits the next crop. The recommendations for N fertilization in an intercropped system are not yet clear. The best yields were obtained in this study with the application of 160 kg ha$^{-1}$ N, which is comparable with the findings of Souza and Soratto [50], who also observed an increase in maize grain yield grown as a relay crop when 120 kg ha$^{-1}$ was applied. These results suggest that intercropping maize with Guinea grass is an alternative to avoid N losses [39], due to its potential for deep soil exploration.

## 5. Conclusions

The results of this study provide important information on the effects of soil acidity alleviation and N supply on soybean and maize root growth in a no-till system. Soil acidity correction and N supply result in better distribution of the soybean and maize root systems in the soil profile, increasing soil exploration, which facilitates water extraction in periods of scarcity and nutrient absorption in deeper layers of the soil, ultimately resulting in higher yields.

The combination of lime with gypsum is an important alternative to increase $Ca^{2+}$ concentrations in the soil profile and improve the distribution of the crop root systems, and N fertilization helps in improving this system, not only in resulting higher maize yields but also improving soybean yields.

These results show the need for and the benefits of N application to maize on the next soybean crop, and they should be considered in recommending soil acidity correction and N fertilization in integrated cropping systems. Future research on this topic should focus on the recovery of the N applied to maize by the next crop.

**Author Contributions:** C.A.R.—experiment planning, funding, writing; M.D.S.—field and lab work, first draft; J.P.d.Q.B.—field and lab work, writing. All authors have read and agreed to the published version of the manuscript.

**Funding:** This work was supported by FAPESP, São Paulo Research Foundation, grant 2017/22134-0.

**Data Availability Statement:** Data will be made available upon reasonable request.

**Conflicts of Interest:** The authors declare no conflict of interest.

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
