# Peer review of "Synergistic Effects of Subsoil Calcium in Conjunction with Nitrogen on the Root Growth and Yields of Maize and Soybeans in a Tropical Cropping System"

_agronomy, doi:10.3390/agronomy13061547_

Round 1

Reviewer 1 Report

The present research article entitled “Synergistic effects of subsoil calcium associated with nitrogen on root growth and yield in a tropical cropping system" submitted by De Souza et al., in Agronomy journal, discussed the impact of interactive effect of subsoil calcium associated with N application on growth and development/ production of Guinea grass in arid cropping system.

 The experiment is well organized/ planned and the results are significant for the agro-farmers. The MS can be acceptable for publication after revision.

Some specific comments are as stated below:

Please revise the MS title as “Synergistic effects of subsoil calcium associated with nitrogen on root growth and yield of Guinea grass (Megathyrsus maximus) in tropical cropping system”

Line 9-22: Please mention the gain or loss (%) values

Line 22: Please incorporate a key message of the study

Material and Method section needs improvement

Result and discussion section is fine.

Improve conclusion section and incorporate future recommendations

Author Response

The experiment is well organized/ planned and the results are significant for the agro-farmers. The MS can be acceptable for publication after revision.

Some specific comments are as stated below:

Please revise the MS title as “Synergistic effects of subsoil calcium associated with nitrogen on root growth and yield of Guinea grass (Megathyrsus maximus) in tropical cropping system”

Sorry, but we studied roots of maize + Guinea grass and soybean, and yields of soybean and maize. Therefore, the suggested title would not be correct.

Line 9-22: Please mention the gain or loss (%) values

edited

Line 22: Please incorporate a key message of the study

Edited

Material and Method section needs improvement

Sorry, we could not see how to improve it.

Result and discussion section is fine.

Improve conclusion section and incorporate future recommendations

edited

Reviewer 2 Report

please open the attached file

the  English Language of manuscript is accepted 

Author Response

The systematic abstract is missing. Introduce the need for study in 1-2 lines. Then please give a clear-cut point problem source as a problem statement that is tackled in the current study. Quantitative data is also essential to support your conclusion. I request that the authors carefully check and rewrite the results part of the abstract. Please provide a percentage increase or decrease in the result part. Also, give a logical reason for selecting the current strategy or treatments. Then provide a definitive conclusion withdrawn through research in a single line.

The abstract was edited

In the introduction section, the author should provide a novelty statement at the end. What new things have authors done or correlated in this research compared to old ones?

Edited

The authors should follow the title in the introduction section, i.e., Synergistic effects of subsoil calcium associated with nitrogen on root growth and yield in a tropical cropping system. Do you consider the topic original or relevant in the field? Does it address a specific gap in the field?

Edited

 In page 4, line 113, change the unit Mg ha-1 to international unit ( ton ha-1 )

Mg ha_1 is an International Unit

In Page 5 line 153, the authors should add the model, country, and manufactured year for the forced-air oven as they did for other tools that have been used in this research study

The year of manufacturing is not important.

The authors should organize the tables and figures well, where, the color of figure is not enough clear (black color and Palatino Linotype font).

We could not understand what the problem would be here. The font is not Platino and there is just a few columns in black.

For results section, the authors should provide Pearson correlation for a better understanding the relationship between the different variables, special, Soil Ca concentration and root, yield and vegetative growth parameters.

We think this would imply in more tables since there are many soil depths, years, N rates, etc, and would add very little to the interpretation and understanding of the results.

I suggest that the authors should add photos for the plant roots in supplementary data, if that possible

There are no photos, since taking photos in the field would damage the plots.

Why the authors focused only on root and yield parameters and left other vegetative growth parameters such as chlorophyll content, fresh and dry weight of plants, leaf area, number of leaves……..etc.

The objective was to study how the association of lime, gypsum and N would affect root growth and yield. In these systems water shortage is an issue for, at least, the maize phase, that is why root growth is so important. W think vegetative parameters would not add to our discussion, because there would be secondary effects.

If the authors did not record the pervious vegetative growth parameters (chlorophyll content, fresh and dry weight of plants, leaf area, number of leaves). It should be covered them in discussion section

It would be speculative.

For conclusion section, the authors should write conclusion separately from the discussion part and they should provide a conclusive conclusion, Add the targeted beneficiary audience who will get benefit from this research. Also, give clear-cut recommendations Give future prospective regarding this research.

We did not understand what the problem would be. Conclusions is item 5, separated from discussion. However, it was edited.

For reference section , the author should follow the format of reference related to Agronomy MDPI and Update the old references (citations) especially numbers 6,7,11,12,10, 12, and 13.

Thank you. Corrected. Frontiers in Plant Science (REF 10) does not show pages.